PROCEEDINGS
OF SCIENCE

# Position-Dependent Mass Quantum systems and ADM formalism

**Davood Momeni** *†

*Department of Physics, College of Science, Sultan Qaboos University , Al Khodh 123 , Muscat, Oman*

*E-mail:* davood@squ.edu.om

The classical Einstein-Hilbert (EH) action for general relativity (GR) is shown to be formally analogous to the classical system with position-dependent mass (PDM) models. The analogy is developed and used to build the covariant classical Hamiltonian as well as defining an alternative phase portrait for GR. The set of associated Hamilton's equations in the phase space is presented as a first order system dual to the Einstein field equations. Following the principles of quantum mechanics,I build a canonical theory for the classical general. A fully consistent quantum Hamiltonian for GR is constructed based on adopting a high dimensional phase space. It is observed that the functional wave equation is timeless. As a direct application, I present an alternative wave equation for quantum cosmology. In comparison to the standard Arnowitt-Deser-Misner(ADM) decomposition and qunatum gravity proposals, I extended my analysis beyond the covariant regime when the metric is decomposed in to the $3+1$ dimensional ADM decomposition. I showed that an equal dimensional phase space can be obtained if one apply ADM decomposed metric.

*International Conference on Holography, String Theory and Discrete Approaches in Hanoi Phenikaa University*
*August 3-8, 2020*
*Hanoi, Vietnam*

---

*Speaker.
†Also at Center for Space Research, North-West University, Mafikeng, South Africa.

                                        

## 1. Introduction

Classical mechanics is Galilean invariant, i.e, time parameter $t$ and position coordinate $q(t)$ are explicitly functions of each other. Since quantum mechanics is Galilean invariant there is no simple way to build a locally Lorentz invariant theory with single particle interpretation (the possible version known as Klein-Gordon has field theoretic realization). In GR, we have difficulty to interpret time as we did in classical mechanics. Furthermore GR is locally Lorentz invariant. The simple reason is that in the context of GR (or any other classical gauge theory for gravity ),time $t$ is just a coordinate and is no longer considered as a parameter (in non-relativistic mechanics and QM the time $t$ is an evolution parameter). As a result ,the analogue to coordinate (in the variational process), the Riemannian metric $g_{ab}(x^c),,a,b = 0...3$ is a function of the all coordinates $x^c = (t,x^A),A = 1,..3$. That makes quantum gravity difficult to build. A way to address quantum gravity is string theory [1] and the other well studied candidate is loop quantum gravity [2]. There is also semi quantum gravity, when we keep the background classical and let fields simply propagate on it. With both of these nice ideas , still we have the problem of "disappearance of time" [3]. If we adopt any of these proposals, it seems that time problem remains unsolved both in the quantum gravity and cosmology [4]. Time problem has a rather long history [5] . There is a simple way to address quantum gravity without time considered in canonical quantization in metric variables:[6]. With all of the above historical backgrounds and many others, we are still looking for a fully covariant canonical quantum theory for gravity which make sense same as we know for usual q mechanics. It is necessary to find an appropriate representation for Lagrangian of the gravity (here GR as the best tested one ).

With a suitable covariant definition of the conjugate momentum we define a Hamiltonian. Furthermore we need to adopt a well defined phase space. In that phase space one can build Poisson brackets easily, and then by replacing the classical bracket with the Dirac bracket, we can find a suitable fully consistent Hamiltonian for quantum GR. Later, one can build an associated (functional) Hilbert space and develop all the concepts of ordinary quantum mechanics systematically. This is a plan to find a successful quantum theory for gravity or as it is known, quantum gravity. During studying non standard classical dynamical systems I found a class of Lagrangian models with second order time derivative of the position $\ddot{q}(t)$(configuration coordinate $q(t)$). It is easy to show that a wide class of such models reduce to the position dependence mass (PDM) models as it was investigated in literature [7]-[9]. It is obviously interesting to show that whether GR reduces to such models. This is what I investigate in this letter. I show that the classical Einstein-Hilbert (EH) Lagrangian reduces to the position-dependent -mass (PDM) model up to a boundary term. Then I adopted the standard quantization scheme for a PDM system and I suggested a fully covariant quantum Hamiltonian for GR. The functional wave equation for the metric proposed naturally and then it was developed for quantum cosmology.

My observations initiated when I studied GR as a classical gauge theory [10]. As everyone knew, GR has a wider class of symmetries provided by the equivalence principle. It respects gauge transformations (any type of arbitrary change in the coordinates, from one frame to the other $x^a \rightarrow \tilde{x}^a$ ) [10]. Consequently GR is considered just a classical gauge theory for gravity. There is also a trivial hidden analogue between GR and classical mechanics (see table I). In GR as a classical dynamical system (but with second order derivative of the position ), if we make an analogous

Riemannian metric $g_{ab}$ with the coordinate $q(t)$ and if we use the spacetime derivatives of the metric $\partial_c g_{ab}$ (which is proportional to the Christoffel symbols $\Gamma_{dab}$ ), instead of the velocity , i.e, the time variation of the coordinate $\dot{q}$, and by adopting a symmetric connection , we can rewrite EH Lagrangian in terms of the metric, first and second derivatives of it. It looks like a classical system in the form of $L(q, \dot{q}, \ddot{q})$ (see TABLE I). In this formal analogy, the classical acceleration term in the classical models under study now $\ddot{q}$ is now replaced with the second derivative for metric i.e, $\partial_e \Gamma_{dab}$. Integration part by part from this suitable representation of the EH Lagrangian reduces it to a PDM system where we will need to define a super mass tensor as a function of the metric instead of the common variable mass function $m(q(t))$ in classical mechanic.

**Table 1:** Analogy between classical mechanics and GR

| Model | Position | first derivative | second derivative | mass |
|---|---|---|---|---|
| Classical PDM | $q(t)$ | $\dot{q}(t)$ | $\ddot{q}(t)$ | scalar $m(q)$ |
| GR | $g_{ab}$ | $\partial_d g_{ab}$ | $\partial_e \partial_d g_{ab}$ | super mass tensor $M^{abldeh}$ given in eq.(2.6) |

In this letter, I focus on the classical EH action for GR as an analogy to the model investigated in the former above. The notation for Einstein-Hilbert action is

$$S_{EH} = \int d^4x \mathscr{L}_{GR} \qquad (1.1)$$

The key idea is to realize GR as a classical dynamical system with second order derivative. The Einstein field equations are derived as a standard variational problem subjected to a set of appropriate boundary conditions. Since the idea of GR is to find the best geometry for a given source of matter fields, it is formally equivalent to the classical mechanics. By combining all these similarities I end up to an equivalent representation of EH action as a classical Lagrangian in the form :

$$\mathscr{L}_{GR} = L(|g|, \partial_a |g|, g_{bc}, g^{bc}, \Gamma^a_{bc}, \partial_d \Gamma^a_{bc}). \qquad (1.2)$$

here $|g| \equiv \det(g_{\mu\nu})$ is determinant of the metric tensor, and $\partial_\beta$ is coordinate derivative. We can write the above Lagrangian formally in a more compact form as

$$\mathscr{L}_{GR} = \mathscr{L}_{GR}(|g|, \partial|g|, g, \overline{g}, \partial g, \partial \overline{g}). \qquad (1.3)$$

here we abbreviated by $g \equiv g_{ab}$ , $\overline{g} \equiv g^{ab}$ , $|g| \equiv \det(g_{ab}) = \frac{1}{\det(\overline{g})}, \frac{1}{2}\overline{g}.\partial g \equiv \Gamma^a_{bc}$ , $\partial g = \Gamma_{bla} \equiv g_{bl,a} + g_{la,b} - g_{ba,l}$. We adopt metricity condition $\nabla_a g^{bc} = |g|^{-1/2}\partial_a(|g|^{1/2}g^{bc}) \equiv |g|^{-1/2}\partial(|g|^{1/2}\overline{g}) = 0$, $\partial.\overline{g} = -\overline{g}.\partial g.\overline{g}$. The plan of this letter is as following: In Sec. (2) I showed that GR Lagrangian reduces to a PDM fully classical system with a super mass tensor of rank six. In Sec. (3) I built a consistent super phase space as well as a set of Poisson brackets. As an attempt to break the complexity of the field equations, I show that gravitational field equations reduced top a set of first order Hamilton's equations. In Sec. (4) I define quantum Hamiltonian simply by replacing the classical brackets with Dirac brackets. Functional wave equations are proposed and by solving them we can obtain generic wave function for a fully canonical quantized Riemannian metric. As a concrete example,in Sec. (5) I solve functional wave equation for a cosmological background. Some asymptotic solutions are presented. The last section is devoted to summarize results.

## 2. Super Mass Tensor for GR as PDM classical system

We adopt the conversion of indices as [11]. The EH action for GR In units $16\pi G \equiv 1$ is

$$S_{EH} = \int_{\mathscr{M}} d^4x \sqrt{g} R \equiv \int_{\mathscr{M}} d^4x \mathscr{L}_{GR}. \tag{2.1}$$

The Ricci scalar $R$ is composed of the metric and its first and second derivatives. The first aim is to express the integrand (Lagrange density $\mathscr{L}_{GR}$) is as the form from which PDM kinetic term is obvious. We note that the Lagrangian density eq. (2.2) is a purely kinetic form, with a PDM effective mass. This adequate representation can be obtained from the definition of $\Gamma_{bla}$, this will be clear if we rewrite the Lagrangian in following equivalent form(note that the Lagrangian enjoys an exchange of indices symmetry $a \to d$ in the first two terms),in action presented in eq.(1.3) one can eliminate the second derivative term $\partial_{de} g_{ab}$ simply by integrating by part and using the metricity condition $\nabla_a g^{bc} = 0$, by taking into the account all the above requirements a possible equivalent form for Lagrangian of the GR is given by:

$$\mathscr{L}_{GR} = \frac{1}{2}\sqrt{|g|}\left(g^{al}g^{be}\Gamma_{bla}\partial^h g_{eh} + g^{be}g^{dh}\Gamma_{bld}\partial^l g_{eh} + \frac{1}{2}g^{al}g^{bd}g^{te}\Gamma_{tld}\Gamma_{bea} - \frac{1}{2}g^{al}g^{bd}g^{te}\Gamma_{tla}\Gamma_{bed}\right) \tag{2.2}$$

and $S_{EH} = \int d^4x \mathscr{L}_{GR} + B.T$ here by $B.T$ we mean boundary term defined as

$$B.T = \int_{\partial\mathscr{M}} \sqrt{|h_{AB}|} h^{BD} h^{AL} \Gamma_{BLA}|_{x^D = constant} + \int_{\partial\mathscr{M}} \sqrt{|h_{AB}|} h^{BD} h^{AL} \Gamma_{BLD}|_{x^A = constant}. \tag{2.3}$$

We can re express the above GR Lagrangian in our convenient notations as

$$\mathscr{L}_{GR} = \frac{1}{2}\sqrt{|g|}\left(\overline{g}.\partial g.\overline{g}.\overline{\partial}g + \overline{g}.\overline{g}.\partial g.\overline{\partial}g + \frac{1}{2}\overline{g}.\overline{g}.\overline{g}.\partial g.\partial g - \frac{1}{2}\overline{g}.\partial g.\overline{g}.\overline{g}.\partial g\right). \tag{2.4}$$

Note that by "." we mean tensor product(we adopt Einstein summation rule). From the above representation we can realize $\{g, \overline{g}\}$ as two fields , in analogy to the Dirac Lagrangian where the fermionic pairs $\psi, \overline{\psi}$ appeared . The difference here is due to the fact that the pair of objects $g, \overline{g}$ depend on each other as we know $g.\overline{g} = \delta$, the Kronecker delta, however in the Dirac Lagrangian the norm $\overline{\psi}\psi \neq I$. In our program we wont use this duality and we will focus on the coordinates representation of the GR Lagrangian, i.e, eq.(2.2) . If we substitute the definition of Gamma terms and combine the theory, we obtain the final form for the Lagrangian as a PDM system for coordinate $g_{ab}$(or as a tensor version for k-essence [12]):

$$\mathscr{L}_{GR} = \frac{1}{2}\sqrt{|g|} M^{abldeh} \partial_a g_{bl} \partial_d g_{eh}. \tag{2.5}$$

here $M^{abldeh} = |g|^{-1/2}\frac{\partial^2 \mathscr{L}_{GR}}{\partial(\partial_a g_{bl})\partial(\partial_d g_{eh})}$ is defined as super mass tensor. An alternative form for (2.5) is $\mathscr{L}_{GR} = \frac{\sqrt{|g|}}{2}\overline{M}\partial g \partial g$. It is equivalent to the classical Lagrangian of PDM systems $L = \frac{1}{2}M(q)\dot{q}^2$ for one dimensional, position dependent mechanical system. As we expected in GR, the mass term transformed to a higher order (here rank six) tensor. The explicit form for the super mass tensor is expressed as following:

$$|g|^{1/2}\overline{M} = |g|^{1/2}M^{a_1 b_1 l_1 d_1 e_1 h_1} = \frac{1}{4}g^{al}\,g^{bd}\,g^{te}\times \tag{2.6}$$

$$\left(\delta_a^{a_1}\,\delta_b^{b_1}\,\delta_e^{l_1} + \delta_a^{b_1}\,\delta_b^{a_1}\,\delta_e^{l_1} - \delta_a^{l_1}\,\delta_b^{b_1}\,\delta_e^{a_1}\right)\left(\delta_d^{d_1}\,\delta_l^{h_1}\,\delta_t^{e_1} - \delta_d^{h_1}\,\delta_l^{d_1}\,\delta_t^{e_1} + \delta_d^{e_1}\,\delta_l^{h_1}\,\delta_t^{d_1}\right)$$

$$-\frac{1}{4}g^{al}\,g^{bd}\,g^{te}\left(\delta_b^{a_1}\,\delta_d^{b_1}\,\delta_e^{l_1} + \delta_b^{b_1}\,\delta_d^{a_1}\,\delta_e^{l_1} - \delta_b^{b_1}\,\delta_d^{l_1}\,\delta_e^{a_1}\right)\left(\delta_a^{d_1}\,\delta_l^{h_1}\,\delta_t^{e_1} - \delta_a^{h_1}\,\delta_l^{d_1}\,\delta_t^{e_1} + \delta_a^{e_1}\,\delta_l^{h_1}\,\delta_t^{d_1}\right)$$

$$+\frac{1}{4}g^{al}\,g^{bd}\,g^{te}\left(\delta_a^{d_1}\,\delta_b^{e_1}\,\delta_e^{h_1} + \delta_a^{e_1}\,\delta_b^{d_1}\,\delta_e^{h_1} - \delta_a^{h_1}\,\delta_b^{e_1}\,\delta_e^{d_1}\right)\left(\delta_d^{a_1}\,\delta_l^{l_1}\,\delta_t^{b_1} - \delta_d^{l_1}\,\delta_l^{a_1}\,\delta_t^{b_1} + \delta_d^{b_1}\,\delta_l^{l_1}\,\delta_t^{a_1}\right))$$

$$-\frac{1}{4}g^{al}\,g^{bd}\,g^{te}\left(\delta_b^{d_1}\,\delta_d^{e_1}\,\delta_e^{h_1} + \delta_b^{e_1}\,\delta_d^{d_1}\,\delta_e^{h_1} - \delta_b^{e_1}\,\delta_d^{h_1}\,\delta_e^{d_1}\right)\left(\delta_a^{a_1}\,\delta_l^{l_1}\,\delta_t^{b_1} - \delta_a^{l_1}\,\delta_l^{a_1}\,\delta_t^{b_1} + \delta_a^{b_1}\,\delta_l^{l_1}\,\delta_t^{a_1}\right))$$

$$+g^{al}\,\delta_a^{d_1}\,g^{be}\,g^{dh}\,\delta_e^{e_1}\,\delta_h^{h_1}\left(\delta_b^{b_1}\,\delta_d^{a_1}\,\delta_l^{l_1} + \delta_b^{a_1}\,\delta_d^{l_1}\,\delta_l^{b_1} - \delta_b^{b_1}\,\delta_d^{l_1}\,\delta_l^{a_1}\right)$$

$$+g^{al}\,g^{be}\,g^{dh}\,\delta_d^{d_1}\,\delta_e^{e_1}\,\delta_h^{h_1}\left(\delta_a^{a_1}\,\delta_b^{b_1}\,\delta_l^{l_1} + \delta_a^{l_1}\,\delta_b^{a_1}\,\delta_l^{b_1} - \delta_a^{l_1}\,\delta_b^{b_1}\,\delta_l^{a_1}\right)$$

$$+g^{al}\,\delta_a^{a_1}\,g^{be}\,g^{dh}\,\delta_e^{b_1}\,\delta_h^{l_1}\left(\delta_b^{e_1}\,\delta_d^{d_1}\,\delta_l^{h_1} + \delta_b^{d_1}\,\delta_d^{h_1}\,\delta_l^{e_1} - \delta_b^{e_1}\,\delta_d^{h_1}\,\delta_l^{d_1}\right)$$

$$+g^{al}\,g^{be}\,g^{dh}\,\delta_d^{a_1}\,\delta_e^{b_1}\,\delta_h^{l_1}\left(\delta_a^{d_1}\,\delta_b^{e_1}\,\delta_l^{h_1} + \delta_a^{h_1}\,\delta_b^{d_1}\,\delta_l^{e_1} - \delta_a^{h_1}\,\delta_b^{e_1}\,\delta_l^{d_1}\right).$$

Having the Lagrangian of GR given in eq. (2.5), one can define a canonical pair of position conjugate momentum $(g,\overline{p})$ and construct a phase space. This is what we are going to do in next section.

## 3. Super phase space

The phase space description of the classical model presented in eq.(2.5) is very straightfor-wardly done, by defining the super conjugate momentum tensor is

$$p^{rst} = \frac{\partial\mathscr{L}_{GR}}{\partial(\partial_r g_{st})} = \frac{\sqrt{g}}{2}\left(M^{rstdeh}\partial_d g_{eh} + M^{ablrst}\partial_a g_{bl}\right). \tag{3.1}$$

Note that the mass tensor $M^{rstdeh}\partial_d g_{eh} = M^{ablrst}\partial_a g_{bl}$. A possible classical Hamiltonian will be

$$\mathscr{H}_{GR} = \frac{1}{2\sqrt{|g|}}M^{abldeh}M_{rstabl}p^{rst}M_{uvwdeh}p^{uvw}. \tag{3.2}$$

A possible Poisson's bracket $\{F,G\}_{P.B}$ adopted to this system is:

$$\{F(g_{mn},p^{stu}),G(g_{mn},p^{stu})\}_{P.B} = \sum\left(\frac{\partial F}{\partial g_{ab}}\frac{\partial G}{\partial p^{rst}} - \frac{\partial F}{\partial p^{rst}}\frac{\partial G}{\partial g_{ab}}\right)_{..} \tag{3.3}$$

or in our notation it simplifies to the following expression

$$\{F(g,\overline{p}),G(g,\overline{p})\}_{P.B} = \sum\left(\frac{\partial F}{\partial g}\frac{\partial G}{\partial \overline{p}} - \frac{\partial F}{\partial \overline{p}}\frac{\partial G}{\partial g}\right). \tag{3.4}$$

and specifically for our super phase coordinates $(g_{ab},p^{rst})$, I I postulate that

$$\{g_{ab},p^{rst}\}_{P.B} = c^r\delta_{ab}^{rs}. \tag{3.5}$$

Here $\delta_{ab}^{rst}$ is the generalized Kronecker defined as [13]

$$\delta_{ab}^{rst} = 2! \delta_{[a}^s \delta_{b]}^t \tag{3.6}$$

In the above Poisson's bracket, with structure constants $c^r$ provide a classical minimal volume for super phase space (zero for Poisson's bracket same objects ). We have now full algebraic structures in the super phase space and canonical Hamiltonian. As a standard procedure, we can write down Hamilton's equations as first order reductions of the Euler-Lagrange equations derived from the Lagrangian given in eq. (2.5)(Einstein field equations). This is one of the main results of this letter and I will address it in the next short section.

## 3.1 Reduction of the Einstein Field Equations (EFE) to Hamilton's equation via covariant Hamiltonian

The set of Hamilton's equations derived from the Hamiltonian (6.3), are defined automatically using the Poisson's bracket are given as following:

$$\partial_a g_{bl} = \{g_{bl}, \mathscr{H}_{GR}\}_{P.B} = \frac{\partial \mathscr{H}_{GR}}{\partial p_{abl}}, \tag{3.7}$$

$$\partial_a p^{abl} = \{p^{abl}, \mathscr{H}_{GR}\}_{P.B} = -\frac{\partial \mathscr{H}_{GR}}{\partial g_{bl}}. \tag{3.8}$$

We explicitly can write this pair of Hamilton's equation given as follows:

$$\frac{2}{\sqrt{g}} \partial_a g_{bl} = M^{a'b'l'deh} M_{rsta'b'l'} M_{uvwdeh} \times \left( \delta_a^u \delta_b^v \delta_l^w p^{rst} + \delta_a^r \delta_b^s \delta_l^t p^{uvw} \right). \tag{3.9}$$

$$-\frac{2}{\sqrt{g}} \partial_a p^{abl} = M^{a'b'l'deh} p^{rst} M_{rsta'b'l'} p^{uvw} \frac{\partial M_{uvwdeh}}{\partial g_{bl}} + M^{a'b'l'deh} p^{rst} M_{uvwdeh} \frac{\partial M_{rsta'b'l'}}{\partial g_{bl}} \tag{3.10}$$

$$+ \frac{\partial M^{a'b'l'deh}}{\partial g_{bl}} p^{rst} M_{rsta'b'l'} p^{uvw} M_{uvwdeh} + M^{a'b'l'deh} \frac{\partial p^{rst}}{\partial g_{bl}} M_{rsta'b'l'} p^{uvw} M_{uvwdeh}$$

$$+ M^{a'b'l'deh} p^{rst} M_{rsta'b'l'} \frac{\partial p^{uvw}}{\partial g_{bl}} M_{uvwdeh}.$$

This set of first order partial differential equations are considered the first phase space alternative to the original gravitational field equations. These equations are considered as important results of my current letter. When we succeed to write a covariance Hamiltonian, the Hamilton's equations are first order version of the Einstein field equations. In my knowledge this is the first time in literature when a first order Hamiltonian version of the gravitational field equations. The set of equations given in (3.7,3.8) are defined when a first first order Hamiltonian version of the field equations for a generic Lorentzian metric. I believe that one can integrate this system as a general non autonomous dynamical system for a given set of the appropriate initial values of the metric and super momentum given as a specific initial position $x_0^a$ (not specific time as is commonly considered as the initial condition in QG literature) .A remarkable observation that the system may possess chaotic behavior and doesn't suffer from Cauchy's problem. We have now the classical Hamiltonian and the set of Poisson brackets. Now we can develop a qunatum version and obtain qunatum Hamiltonian for GR. This will be done in the next section.

## 4. Quantization of GR

In this section, I'm going to define appropriate forms for Dirac brackets simply by defining,

$$\hat{\pi}^{rst} \equiv -i\hbar^r \frac{\partial}{\partial \hat{g}_{st}}, \tag{4.1}$$

$$\left[\hat{g}_{ab}, \hat{\pi}^{rst}\right] = i\hbar^r \delta_{ab}^{st} \tag{4.2}$$

Instead of the usual fundamental reduced planck constant (dirac constant) $\hbar$ we required to define a vector one, the reason is that even the classical phase space spanned by the $\left(g_{ab}, p^{rst}\right)$ one has more degrees of freedom (dof), basically is $10^5 (= 10 \times 10^4)$ dimensional for a Riemannian manifold. The Dirac constant $\hbar$ is proportional to the minimum volume of the phase space $V_0$ defined as

$$\omega_0 = \int D(g_{ab}, p^{rst}). \tag{4.3}$$

where the $D(g_{ab}, p^{rst})$ is a measure for the super phase space and $D(g_{ab}, p^{rst})$ is a covariant volume element. We obviously see that the $\omega_0$ is related to the dof of the system, for example if the system has $f$ numbers of dof, then the minimal volume of the phase space is given as $\hbar^f$ and here $\hbar \propto f^{-1} \log(\omega_0)$, note that in our new formalism $f = 10^5 \gg 1$, as a result the effective $||\hbar^r|| \ll \hbar$.

A remarkable observation is that the super mass tensor $\overline{M} = M^{abldeh}$ is a homogeneous (order 6) of the metric tensor. Using the formalism of quantization for PDM systems the canonical quantized Hamiltonian for GR is:

$$\hat{\mathscr{H}}_{GR}(\hat{g}_{ab}, \frac{\partial}{\partial \hat{g}_{st}}) = -\frac{1}{2} f_{rstuvw}^{1/2} \hbar^r \frac{\partial}{\partial \hat{g}_{st}} \left[ f_{rstuvw}^{1/2} \hbar^u \frac{\partial}{\partial \hat{g}_{vw}} \right]. \tag{4.4}$$

here the auxiliary, scaled super mass tensor $f_{rstuvw}$ is

$$f_{rstuvw} \equiv |g|^{-1/4} M^{abldeh} M_{rstabl} M_{uvwdeh}. \tag{4.5}$$

It is adequate to write the quantum Hamiltonian in the following closed form:

$$\hat{\mathscr{H}}_{GR}(\hat{g}_{ab}, \hat{\pi}^{mnp}) = \frac{1}{2} \left[ |g|^{-1/2} \overline{M} M M \right]^{1/2} \overline{\hat{\pi}} \left[ |g|^{-1/2} \overline{M} M M \right]^{1/2} \overline{\hat{\pi}}. \tag{4.6}$$

where $\overline{p}$ is contravariant component of the super momentum $p$, etc. The above quantization of Hamiltonian is covariant since we didn't specify time $t$ from the other spatial coordinates $x^A$. The model is considered as a timeless model, i.e, there is no first order time derivative in the final wave equation like $\frac{\partial}{\partial t}$, and the associated functional second order wave equation which is fully locally Lorentz invariant as well as general covariant is expressed as:

$$-\frac{1}{2} f_{rstuvw}^{1/2} \hbar^r \frac{\partial}{\partial \hat{g}_{st}} \left[ f_{rstuvw}^{1/2} \hbar^u \frac{\partial}{\partial \hat{g}_{vw}} \Psi(\hat{g}_{ab}) \right] = E \Psi(\hat{g}_{ab})$$

Note that in our suggested functional wave equation for $\Psi(\hat{g}_{ab})$, we end up by the covariant (no first order derivative) of the functional Hilbert space, furthermore all the physical states are static (i.e., no specific time dependency) and consequently we have a covariant full evolution for our functional. I believe that my model is a subclass of the timeless models of QG. Building qunatum gravity via timeless phase space investigated in the past by some authors mainly recent work [14]. Our approach is completely different and independent from the others. We will study qunatum cosmology as a direct application of our wave equation in the next section.

## 5. Quantum cosmology

In flat, Friedmann-Lemaître-Robertson-Walker (FLRW) model with Lorentzian metric $g_{ab} = diag(1, -a(t)\Sigma_3)$ where $\Sigma_3$ is the unit metric tensor for flat space, in coordinates $x^a = (t, x, y, z)$, the non vanishing elements of the super mass tensor defined in eq.(2.6) are

$$M^{abldeh} = -12a^{-2}\delta^{a0}\delta^{d0}\delta^{BL}\delta^{EH} \tag{5.1}$$

here $B, L, E, H = 1, 2, 3$ and the auxiliary scaled super mass tensor $f_{rstuvw}$

$$f_{rstuvw} = -\frac{3a^{1/2}}{4}\delta_{u0}\delta_{r0}\delta_{VW}\delta_{ST}. \tag{5.2}$$

The functional wave equation reduces to the hypersurfaces $\sigma_3$ coordinates $X^A = (x, y, z)$:

$$\frac{3\hbar_0^2 a^{1/2}}{8}\frac{\partial^2\Psi(\hat{g}_{AB})}{\partial\hat{g}_{SS}\partial g_{VV}} = E\Psi(\hat{g}_{AB}) \tag{5.3}$$

and in the coordinates for FLRW metric it reduces simply to the following ordinary differential equation

$$a\Psi''(a) - \Psi'(a) - \frac{32Ea^{5/2}}{3h_0^2}\Psi(a) = 0 \tag{5.4}$$

Here prime denotes derivative with respect to the $a$. If we know boundary conditions, one can build an orthonormal set of eigenfunctions using the Gram Schmidt process. Furthermore the above single value wave equation can be reduce to a standard second order differential equation for wave function $\Psi(a) = \sqrt{a}\phi(a)$,

$$\phi''(a) - \left(\frac{32Ea^{1/2}}{3h_0^2} + \frac{3}{4a^2}\right)\phi(a) = 0 \tag{5.5}$$

It is hard to find an exact solution for the above wave equation but there are exact solutions for asymptotic regimes:

$$\phi(a) \propto \begin{cases} a^{3/2} & \text{if } a \to 0 \\ \exp[\frac{16\sqrt{2E}}{5\sqrt{3}h_0}a^{5/4}] & \text{if } a \to \infty \end{cases} \tag{5.6}$$

and one can build an exact solution via Poincare's asymptotic technique, i.e, by suggesting $\phi(a) = \zeta(a)a^{3/2}\exp[\frac{16\sqrt{2E}}{5\sqrt{3}h_0}a^{5/4}]$ and $\zeta(a)$ will come as a transcendental (hypergeometric) function(

$$\zeta(a) = \frac{3^{2/5}\hbar^{4/5}e^{-\frac{2}{15}\left(\frac{8\sqrt{6e}a^{5/4}}{\hbar} + 3\right)}}{64\sqrt[5]{5}a}\left(5c_1\Gamma\left(\frac{1}{5}\right)I_{-\frac{4}{5}}\left(\frac{16\sqrt{\frac{2e}{3}}a^{5/4}}{5\hbar}\right) + 32\sqrt[5]{2}c_2\Gamma\left(\frac{4}{5}\right)I_{\frac{4}{5}}\left(\frac{16\sqrt{\frac{2e}{3}}a^{5/4}}{5\hbar}\right)\right) \tag{5.7}$$

here $I_v(y)$ is the modified Bessel function of the first kind

and the eigenvalue $E$ (positive, negative or zero ) can be discrete as well as continuous (bound states for $E < 0$ ). Remarkable is for vanishing energy state, $E = 0$, the generic wave function is given by

$$\Psi(a) = N_0 + Na^2, \ a \in [0, \infty) \tag{5.8}$$

More details about qunatum cosmology will appear in my forthcoming paper in preparation [15].

## 6. Note about ADM decomposition formalism and reduced phase space

Working with an extended phase space with a conjugate momentum with one more index doesn't look very friendly at all, although that is the unique way to define a fully covariant form for the phase space as well as a purely kinetic Lagrangian for GR. If one adopt the ADM decomposition of the metric $g_{ab}$ as follows[16, 17],

$$ds^2 = g_{ab}dx^a dx^b = h_{AB}dx^A dx^B + 2N_A dx^A dx^0 + (-N^2 + h^{AB}N_A N_B)(dx^0)^2 \tag{6.1}$$

here $x^0$ is time, $A, B = 1, 2, 3$ refer to the spatial coordinates and $h_{AB}$ is spatial metric. It is easy to show that the set of the first order Hamilton's equations presented in the previous section reduces to the ADM equations, only if one consider $t$ as dynamical time evolution. Basically if we recall the super conjugate momentum

$$\pi^{rs} = \frac{\partial \mathscr{L}_{GR}}{\partial \dot{g}_{st}} = \frac{\sqrt{g}}{2} \left( M^{0stdeh}\partial_d g_{eh} + M^{abl0st}\partial_a g_{bl} \right). \tag{6.2}$$

Builiding the Hamiltonian in a standard format as

$$\mathscr{H}_{GR}^{ADM} = \frac{1}{2\sqrt{|g|}} M^{abldeh} M_{0stabl}\pi^{st} M_{0vwdeh}\pi^{vw}. \tag{6.3}$$

Briefly I wanna to mention here that although my formalism is worked with covariant derivative without specifying any coordinate as time (so technically is a timeless technique)if one reback to the standard metric decomposition in ADM and use the 0 component of super conjugate momentum, again we can recover ADM Hamiltonian. I emphasis here that my construction was based on purely geometrical quantization of the GR action rather opting a standard time coordinate .

## 7. Final remarks

The canonical covariant quantization which I proposed here is a consistent theory. I started it by basic principles, just by rewriting the GR action in a suitable form the Lagrangian reduced to a purely kinetic theory with position dependence mass term. In this equivalent form of the Lagrangian, gradient of the metric tensor appears as a hypothetical scalar field. With such a simple quadratic Lagrangian, I defined a conjugate momentum corresponding to the metric tensor. Mass term for graviton derived as tensor of rank six . I developed a classical Hamiltonian using the metric and its conjugate momentum. It is remarkable that one can write classical Hamilton's equations for metric and momentum (super phase space coordinates) are analogous to the second order nonlinear Einstein field equations. Later I replaced Poisson's brackets with Moyal(Dirac) and I defined a quantum Hamiltonian for GR. There is no time problem in this formalism because theory is fully covariant from the beginning. As a direct application I investigated qunatum cosmology, i.e and wave function for a homogeneous and isotropic Universe. I showed that wave equation simplifies to a linear second order ordinary differential equation with appropriate asymptotic solutions for very early and late epochs. In my letter I used an integration part by part to reduce GR Lagrangian to a form with first derivatives of the metric. The price is to define two boundary terms on the spatial boundary regions. Those terms vanish in any asymptotic flat(regular) metric. I notice here

that even if we didn't remove second derivative terms using integration by part, it was possible to define a second conjugate momentum $r^{abcd} = \frac{\partial \mathscr{L}_{GR}}{\partial(\partial_a\partial_b g_{cd})}$ corresponding to the second derivative of the metric $\partial_a\partial_b g_{cd}$. If I impose a Bianchi identify between $g_{ab}, p^{rst}, r^{abcd}$, it is possible to fix this new momentum in terms of the other one and the metric. A suitable Legendre transformation from the GR Lagrangian

$$\mathscr{H}_{GR} = p^{rst}\partial_r g_{st} + r^{abcd}(\partial_a\partial_b g_{cd}) - \mathscr{L}_{GR} \tag{7.1}$$

with the Bianchi identity,

$$\{g_{ab}, \{p^{cde}, r^{fghi}\}_{P.B}\}_{P.B} + \{p^{cde}, \{r^{fghi}, g_{ab}\}_{P.B}\}_{P.B} + \{r^{fghi}, \{g_{ab}, p^{cde}\}_{P.B}\}_{P.B} = 0. \tag{7.2}$$

I obtain a standard Hamiltonian without this new higher order momentum. Consequently the method of finding a Hamiltonian based on the super phase spaces which I defined by $g_{ab}, p^{rst}$ in Sec. II and the one with one more conjugate momentum i.e, $g_{ab}, p^{rst}, r^{abcd}$ leads to the same result. I will demonstrate this in a forthcoming longer paper[15].

## 8. Acknowledgments

This work supported by the Internal Grant (IG/SCI/PHYS/20/07) provided by Sultan Qaboos University . The tensor manipulations of this done using WXMAXIMA platform.

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
