# Peer review of "Position-Dependent Mass Quantum systems and ADM formalism"

_SciPost Physics Proceedings_

## Round 1 · Referee Report · Anonymous · 2020-10-21

Report

The manuscript “Position-Dependent Mass Quantum systems and ADM formalism”, by Davood Momeni

tries to present a Hamiltonian formulation of general relativity, inspired by position-dependent mass systems, and then it proposes a quantum version of the theory.

1) The Hamiltonian formulation of general relativity has been discussed extensively in the literature for decades. Some references are given, but many important contributions are missing (one can find them in textbooks or Review papers). In any case, the author should tone down his claims about his findings and their significance, and remove statements such as “this is the first time in literature when a first order Hamiltonian version of the gravitational field equations.”

2) The author’s approach is based on the “super mass tensor”. However, the author defines this quantity as a derivative on the Lagrangian, and hence expression (2.5) is a loop definition.

3) I cannot see how the complicated form of field equations (3.10) can be helpful.

4) The matter sector, which is crucial in GR since it is the source of non-trivial curvature and geometry, is missing from the discussion.

5) The author faces the problem as a simple quantum mechanical problem and not as a quantum field theoretical one, and thus the discussion on renormalizability etc is missing.

6) Solutions (5.7),(5.8) do not have an obvious meaning, and in any case it is strange that the author finds non-trivial structure in the absence of matter.

7) The English of the manuscript need editing.

In summary, a radical modification is needed before I will be able to reconsider the manuscript for publication.

---

## Editorial Decision

resubmitted